# Single-peptide DNA-dependent RNA polymerase homologous to multi-subunit RNA polymerase

David Forrest[1], Katherine James[1], Yulia Yuzenkova[1] & Nikolay Zenkin[1]

Transcription in all living organisms is accomplished by multi-subunit RNA polymerases (msRNAPs). msRNAPs are highly conserved in evolution and invariably share a ∼400 kDa five-subunit catalytic core. Here we characterize a hypothetical ∼100 kDa single-chain protein, YonO, encoded by the SPβ prophage of *Bacillus subtilis*. YonO shares very distant homology with msRNAPs, but no homology with single-subunit polymerases. We show that despite homology to only a few amino acids of msRNAP, and the absence of most of the conserved domains, YonO is a highly processive DNA-dependent RNA polymerase. We demonstrate that YonO is a *bona fide* RNAP of the SPβ bacteriophage that specifically transcribes its late genes, and thus represents a novel type of bacteriophage RNAPs. YonO and related proteins present in various bacteria and bacteriophages have diverged from msRNAPs before the Last Universal Common Ancestor, and, thus, may resemble the single-subunit ancestor of all msRNAPs.

[1] Centre for Bacterial Cell Biology, Institute for Cell and Molecular Bioscience, Newcastle University, Baddiley-Clark Building, Richardson Road, Newcastle upon Tyne NE2 4AX, UK. Correspondence and requests for materials should be addressed to Y.Y. (email: y.yuzenkova@ncl.ac.uk) or to N.Z. (email: n.zenkin@ncl.ac.uk).

The msRNAP evolved before the Last Universal Common Ancestor (that is, the divergence of bacteria and archaea/eukaryotes; LUCA), and already had a 5-subunit ($2\alpha$, $\beta$, $\beta'$, $\omega$ in bacterial nomenclature) catalytic core including most of the domains that are believed to be essential for its functions[1–4]. Evolutionarily, msRNAPs are unrelated to DNA polymerases or to the known viral single-subunit RNAPs, for instance T7 RNAP[5,6]. However, bioinformatic and, later, structural analyses revealed a group of single-subunit proteins that are very distant relatives of msRNAP and which must have diverged from the msRNAP branch far before the LUCA. This group includes the eukaryotic RNA-dependent RNAPs involved in post-transcriptional gene silencing, and several hypothetical proteins present in the prophages of some firmicutes, and in the main genomes of many firmicutes and cyanobacteria[3,7–9]. These latter proteins are referred to here as 'YonO-like' after the predicted protein YonO of *B. subtilis* prophage SPβ.

The only homology of eukaryotic RNA-dependent RNAP and YonO-like proteins to msRNAP lies within the two double-psi-β-barrel domains of the $\beta$ and $\beta'$ subunits (Fig. 1a). Structural predictions and structural analysis have indicated the presence of these domains in YonO-like proteins and eukaryotic RNA-dependent RNAP, respectively[3,7–10]. These domains carry several absolutely conserved amino acids that participate in binding of the catalytic metal and incoming nucleotides, and are believed to be the most ancient domains of msRNAPs[3,7–9] (Fig. 1a). Clear, though very distant, homology to msRNAPs rules out the possibility of a relationship between either eukaryotic RNA-dependent RNAP or YonO and the known single-subunit polymerases.

No sequence homology to msRNAPs beyond the few conserved amino acids, mentioned above, is detectable in YonO. Furthermore, being approximately four times smaller than the conserved catalytic core of msRNAPs, YonO is likely to lack most of the domains that are essential for msRNAP's function[1]. The activity and function of YonO or related hypothetical proteins are unknown.

Here, we studied YonO, both *in vivo* and *in vitro*, and show that it is a new type of specific and highly processive DNA-dependent RNAP and a *bona fide* RNAP of the SPβ bacteriophage of *B. subtilis*.

## Results

**YonO is a DNA-dependent RNA polymerase.** The extremely low homology with msRNAPs and its size (Fig. 1a) suggest that YonO may have lost its nucleic acid polymerization activity in the course of the billions of years following divergence from the msRNAP branch. To test if YonO functions as an RNAP, we used assembled elongation complexes (scheme in Fig. 1b), a technique previously used to investigate various msRNAPs[1,11]. Elongation complexes (ECs) are assembled with purified RNAP, synthetic RNA and template DNA strands, followed by the addition of the non-template DNA strand (scheme in Fig. 1b), and are indistinguishable from native ECs obtained by transcription from a promoter (for DNA and RNA sequences, see the figures and Supplementary Tables). ECs are then immobilized on $Ni^{2+}$-NTA-agarose beads via a 6xHis tag on the RNAP and their stability can then be tested by the retention of radiolabelled RNA on the beads after extensive washing. We found that YonO readily formed stable complexes with the RNA–DNA hybrid, a property expected from RNAP (Fig. 1b). Addition of the non-template strand that was fully complementary to the template led to a destruction of a proportion of the complexes (an effect observed during assembly of ECs with msRNAP); however, a large portion of the ECs remained stable. These ECs also withstood high salt (1 M KCl) washings, similarly to ECs formed by msRNAPs[11] (Supplementary Fig. 1a). Formation of a stable EC by YonO required an 8 bp long RNA–DNA hybrid,

similar to the requirements of msRNAPs (Supplementary Fig. 1b)[11,12]. These consistencies suggest similarities in the organization of the YonO and msRNAP ECs, despite YonO lacking most of domains essential for EC formation by msRNAP[1].

The addition of nucleotide triphosphates (NTPs) to the YonO EC resulted in efficient extension of the RNA to the end of the DNA template (Fig. 1b), indicating that, despite very low homology with msRNAPs, YonO is indeed a template-dependent RNAP. The catalytic aspartate triad of msRNAPs is among the few amino acids conserved in YonO (Fig. 1a). To confirm that this aspartate triad is involved in catalysis by YonO, we prepared a mutant enzyme, $YonO^{3D>3N}$, in which these aspartates were substituted by asparagines. $YonO^{3D>3N}$ formed stable ECs, but was completely inactive in RNA extension, even with a high concentration of NTPs and prolonged incubation (Fig. 1c). This result indicates that the aspartate triad of YonO is functionally homologous to that of msRNAPs.

During elongation, msRNAPs induce disengagement of the RNA from the template DNA at the rear end of the RNA–DNA hybrid[13]. This process maintains the length of the RNA–DNA hybrid, allows the reformation of the upstream DNA duplex and preserves the geometry of the EC during elongation. The failure to do so would result in the formation of an extended RNA–DNA hybrid, which would preclude the release of the transcript from the template DNA. To ascertain if transcription by YonO proceeds with the disengagement of RNA from the DNA template, we chased the ECs formed with YonO or *Escherichia coli* msRNAP to the end of template in the presence of all NTPs. Both RNAPs and the transcripts remained bound at the end of the DNA (Supplementary Fig. 1c), a phenomenon observed previously with msRNAPs[14]. To check if the upstream part of the transcript was annealed to the template, the complexes were treated with RNase H that cleaves RNA within an RNA–DNA hybrid (Supplementary Fig. 1c, lanes 6 and 11). As can be seen from Supplementary Fig. 1c, the transcripts were resistant to RNase H in both YonO and *E. coli* msRNAP reactions, indicating that, similarly to msRNAPs, YonO facilitates the disengagement of RNA from the template strand during elongation.

We analysed the specificity of YonO towards the nature of the template, the transcript and the substrates. As can be seen from Fig. 1d, YonO is a strict (even stricter than the *E. coli* msRNAP used as a control) DNA-dependent RNA polymerase, as it strongly preferred NTPs to dNTPs, DNA to RNA template, and RNA to DNA transcript. msRNAPs can use $Mn^{2+}$ instead of $Mg^{2+}$ as catalytic metal ions, which may relax the specificity towards some reactants. We tested if YonO can use $Mn^{2+}$ for catalysis, and if $Mn^{2+}$ could change YonO's specificity towards template and/or transcript and/or substrates. As seen from Supplementary Fig. 1d, the specificity of YonO in the presence of $Mn^{2+}$ was essentially the same as in the presence of $Mg^{2+}$ (Fig. 1d), with the exception of the extension of the DNA transcript on the DNA template with ribo NTPs, where $Mn^{2+}$ stimulated extension relative to $Mg^{2+}$. It is possible that $Mn^{2+}$ increases the affinity of YonO towards a DNA–DNA 'hybrid'. It is also possible that $Mn^{2+}$ may help melting of the DNA–DNA duplex at the rear edge of the EC, which otherwise would block propagation of the EC[13] (compare lanes 32 and 34 in Supplementary Fig. 1d). However, a requirement for the extension of a DNA primer on a DNA template with ribo NTPs is unlikely to happen *in vivo*.

**YonO is more processive but less accurate than msRNAP.** We compared the kinetics of transcription by YonO and bacterial msRNAP from *E. coli* (Fig. 1e). YonO rapidly extends RNA with little pausing at a low (1 µM) NTP concentration, in contrast to

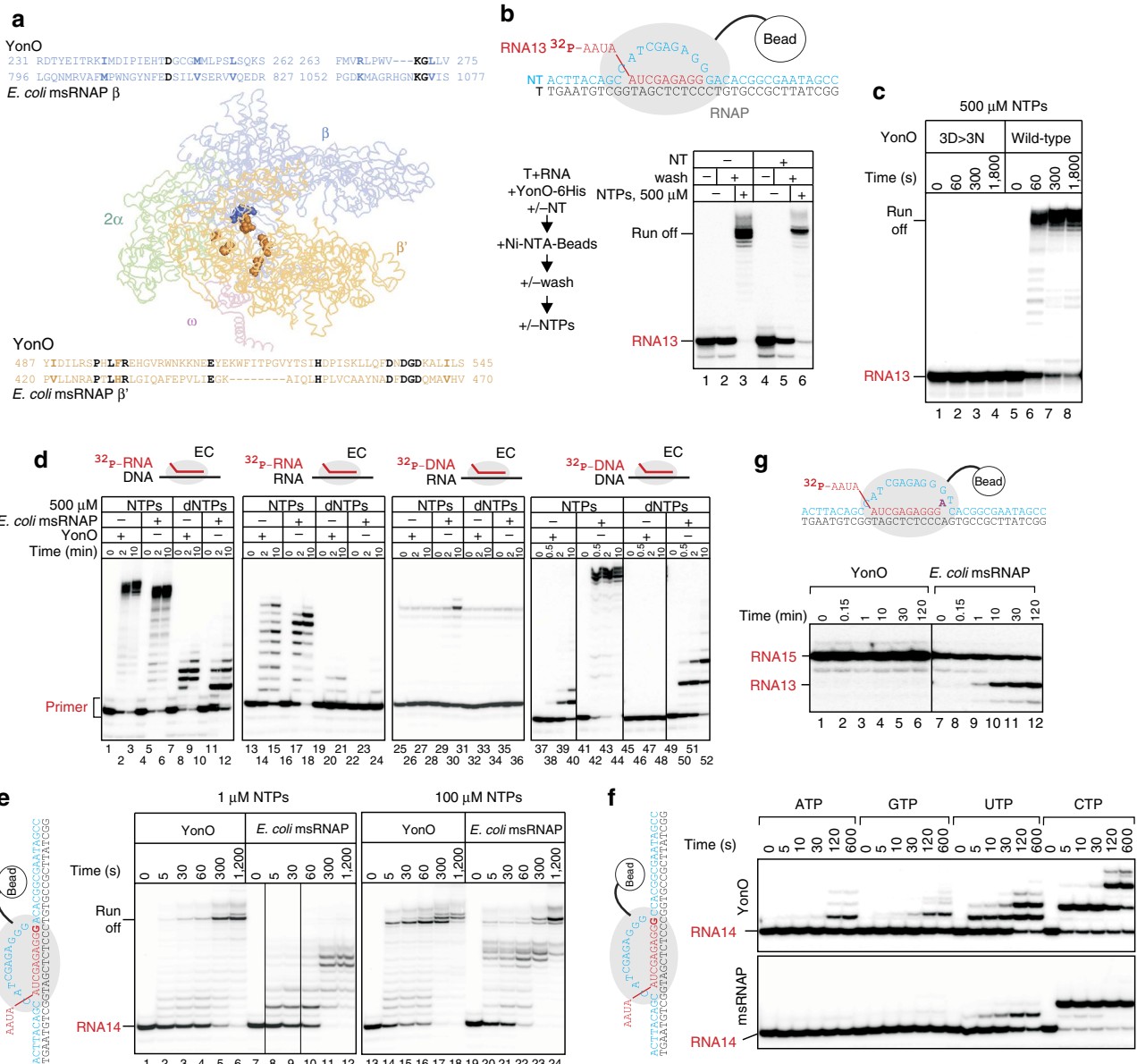

**Figure 1 | YonO is an RNAP. (a)** Detectible homology of YonO and msRNAP is marked in the alignment (identical amino acids—black, conserved substitutions—bold). Identical amino acids are shown on the crystal structure of *E. coli* msRNAP (pdbID 4IGC) as spheres. **(b)** YonO forms stable active elongation complexes. Scheme of the experiment and the sequences of nucleic acids used are shown next to the gel (here and after, for sequences, see Supplementary Table 2). Partial destruction of ECs upon addition of access of the non-template DNA strand (lane 5) is also commonly observed for msRNAP. **(c)** RNA extension in ECs (as in panel (**b**)) formed by wild-type YonO and mutant YonO carrying asparagine substitutions of the aspartate triad homologous to the absolutely conserved catalytic aspartate triad of msRNAPs. **(d)** Specificity of YonO and msRNAP to RNA versus DNA as primers and templates (see Supplementary Table 2 for sequences), and NTP versus dNTP as substrates. A higher molecular weight band in the third panel that coincides with the extension product is a contaminant in the preparation of the DNA primer. **(e)** Kinetics of DNA-dependent RNA polymerization by YonO and msRNAP in the presence of all NTPs. RNA in the EC was labelled at the 3′ end by incorporation of α-[32P]GMP, shown in bold in the scheme next to the gel. **(f)** YonO is more error prone than msRNAPs. Kinetics of misincorporation by YonO and *E. coli* msRNAP. RNA was labelled as in panel (**e**). Note that RNAs of different sequences are well resolved in this Urea-PAGE excluding the possibility that the extension is caused by the contamination with correct NTPs. **(g)** Kinetics of RNA hydrolysis by YonO and msRNAP (see also Supplementary Fig. 1f).

msRNAP, which paused strongly at multiple positions. The higher processivity of YonO was not due to a higher affinity to NTPs, as the same propensity was observed at a high (100 μM) NTP concentration. In addition, the $K_m$ for NTPs appeared to be similar for YonO and *E. coli* msRNAP; $K_m$[ATP] (measured at 10 °C to allow manual measurements) was $1.8 \pm 0.7$ and $0.6 \pm 0.2$ μM, respectively.

We noted that, although the rate of NTP addition by YonO strongly depended upon the base of the template strand, YonO was more error prone than bacterial RNAP (Fig. 1f and Supplementary Fig. 1e). Indeed, we found that affinity for the incorrect NTP was much higher for YonO than for *E. coli* RNAP; $K_m$[incorrectGTP] was $126 \pm 41$ and $458 \pm 23$ μM, respectively. The major determinant of transcription accuracy by msRNAPs is

the catalytic domain Trigger Loop[15,16]. A part of the mechanism for discrimination against incorrect NTPs involves the Trigger Loop competing with them in the active centre[15]. The higher affinity to incorrect NTPs of YonO, thus, may reflect a simpler organization of its active centre, in particular, the absence of homology to the Trigger Loop.

The msRNAP active centre can hydrolyse phosphodiester bonds of the transcript, the reaction required to proofread misincorporation events[17,18] (Fig. 1g). We, however, could not detect any hydrolytic activity by YonO, even when using an EC that mimics a misincorporation event and is stabilized in the conformation most suitable for efficient hydrolysis (Fig. 1g). The deficiency in hydrolysis by YonO was unlikely due to the higher pKa of the reaction, because no hydrolysis was observed at a higher pH (up to pH 10) and during prolonged incubation (Supplementary Fig. 1f). This result further suggests differences in the active centre organization, or, alternatively, an inability of the YonO EC to adopt the backtracked conformation. YonO, however, could perform pyrophosphorolysis (reversal of NMP addition)—another reaction in the reverse direction to RNA extension (Supplementary Fig. 1g).

**YonO is a *bona fide* RNAP of *B. subtilis* prophage SPβ**. The above *in vitro* results show that YonO is a functional and processive DNA-dependent RNA polymerase. *In vivo* we found that YonO is expressed upon the induction of the SPβ prophage with mitomycin C, a common way to induce the lytic cycle of SPβ prophage (Fig. 2a). This observation suggested that YonO might be a *bona fide* bacteriophage RNAP that transcribes a set of the phage genes and is required for its development. To test this hypothesis we deleted the gene encoding YonO within the SPβ prophage. Since mitomycin C also induces production of non-SPβ lytic enzymes in addition to SPβ[19,20], mitomycin C treatment leads to cell lysis of both wild type and ΔyonO strains. Therefore, we tested if the mutant prophage could still produce active bacteriophage particles by analysing the ability of the mitomycin C induced lysates to re-infect *B. subtilis* (strain CU1065 sensitive to SPβ)[21]. As seen from Fig. 2b, addition of the wild-type *B. subtilis* lysate, formed upon induction of lytic cycle, to CU1065 cells, resulted in the formation of plaques. In contrast, lysates of ΔSPβ (a strain lacking SPβ used as a control) and of ΔyonO both failed to form plaques, suggesting that YonO is indeed essential for bacteriophage development.

To directly observe transcription by YonO *in vivo*, and to reveal the genes transcribed by it, we performed transcriptome sequencing in wild type and ΔyonO strains with or without SPβ prophage induction. Upon treatment of the wild-type strain with mitomycin C, 148 out of 184 SPβ prophage genes were upregulated (green dots in Fig. 2c), consistent with the induction of the lytic cycle, as well as up- and downregulation of 579 host genes (shown as pink dots in Fig. 2c). However, of these 148 SPβ genes, 37 (labelled pink dots in Fig. 2d) remained silent upon phage induction in the ΔyonO strain, indicating that they are transcribed by YonO. These genes include structural and lysis genes, which together make up the cluster of late genes[22] (Supplementary Fig. 2). This result is consistent with the strategies of some bacteriophages, where the late genes are transcribed by the phage's own RNAP. In addition to 37 SPβ genes, only 16 non-SPβ genes (pink dots without labels in Fig. 2d; blue dots represent the rest of the genome) exhibited small but statistically significant differences in expression, suggesting that YonO does not have an impact on host expression. Enrichment for 5′ triphosphorylated transcripts prior to sequencing and bioinformatic analysis (RNA was

treated with Terminator 5′-Phosphate-Dependent Exonuclease, which degrades RNAs containing a 5′-monophosphate, thereby enriching for transcripts containing 5′-triphosphates) suggested that YonO transcribes the late genes as one operon, initiating transcription upstream of the yonK gene (and an unannotated non-coding RNA; position 2225998 of *B. subtilis* 168 genome). We confirmed this transcription start site by *in vivo* primer extension (Fig. 2e and Supplementary Fig. 2). Further investigation will be required to determine the promoter sequence recognized by YonO, and whether YonO can recognize promoters on its own or requires additional host or bacteriophage factors.

## Discussion

The principal discovery of our work is that YonO, and potentially its homologues in other prophages, represent a new type of bacteriophage RNAP. Intriguingly, YonO-like proteins were also found in the main genomes of many other firmicutes, cyanobacteria, and in two members of the CFB group (*Bacteroides capillosus* and *Bacteroides pectinophilus*)[7]. It is tempting to speculate that these polymerases may serve to transcribe specific sets of genes in these bacteria in response to environmental changes or stresses.

As YonO has been predicted to have diverged from msRNAPs before the LUCA, structural and functional analysis of YonO will bring insights into the emergence and evolution of msRNAPs. Apart from several amino acids that participate in catalytic $Mg^{2+}$ and substrates binding, there is no sequence homology between active centres of YonO and msRNAPs. Notably, there is no detectable homology to an essential catalytic domain of msRNAPs, the Trigger Loop. The Trigger Loop participates in the catalysis of all reactions performed by msRNAP and is a major determinant of transcription fidelity and proofreading[15,23–26]. It is therefore surprising that YonO catalyses phosphodiester bond synthesis as efficiently as msRNAP. The lack of the homology to the Trigger Loop may however explain a lower accuracy of RNA synthesis and the lack of proofreading (RNA cleavage) activity by YonO. It is possible that the lower overall fidelity could be beneficial for the bacteriophage survival. The previously known single-subunit RNAPs of bacteriophages, such as T7 RNAP, are more processive than msRNAPs. Though YonO cannot be structurally compared to them, the bypass of the pausing signals might be an example of functional convergence of polymerases that need to synthesize a large number of gene copies with minimal delays.

The single-subunit YonO is approximately four times smaller than the core of msRNAP. However, we show that YonO is able to form a stable elongation complex with similar characteristics to that of msRNAPs. The finding suggests that the organization of nucleic acids in the transcription elongation complex was defined early in evolution, before the emergence of multiple domains of msRNAP that are involved in contacts with nucleic acids. We suggest that these conserved domains of msRNAPs have evolved to fine-tune regulation of transcription, rather than to robustly perform accurate template-dependent RNA synthesis per se. For instance, the β flap domain interacts with upstream RNA structures and is required for control of hairpin-dependent transcription pausing and termination[27]. The β′ lid helps to separate RNA from the DNA template at the rear of the RNA–DNA hybrid[13] and prevents the formation of the R-loop that may interfere with replication. The β′ zipper is required at particular classes of promoters and may influence transcription pausing[28]. The β′ rudder clamps onto the RNA–DNA hybrid to stabilize the EC in high salt conditions[29] and may also participate in the response to transcription

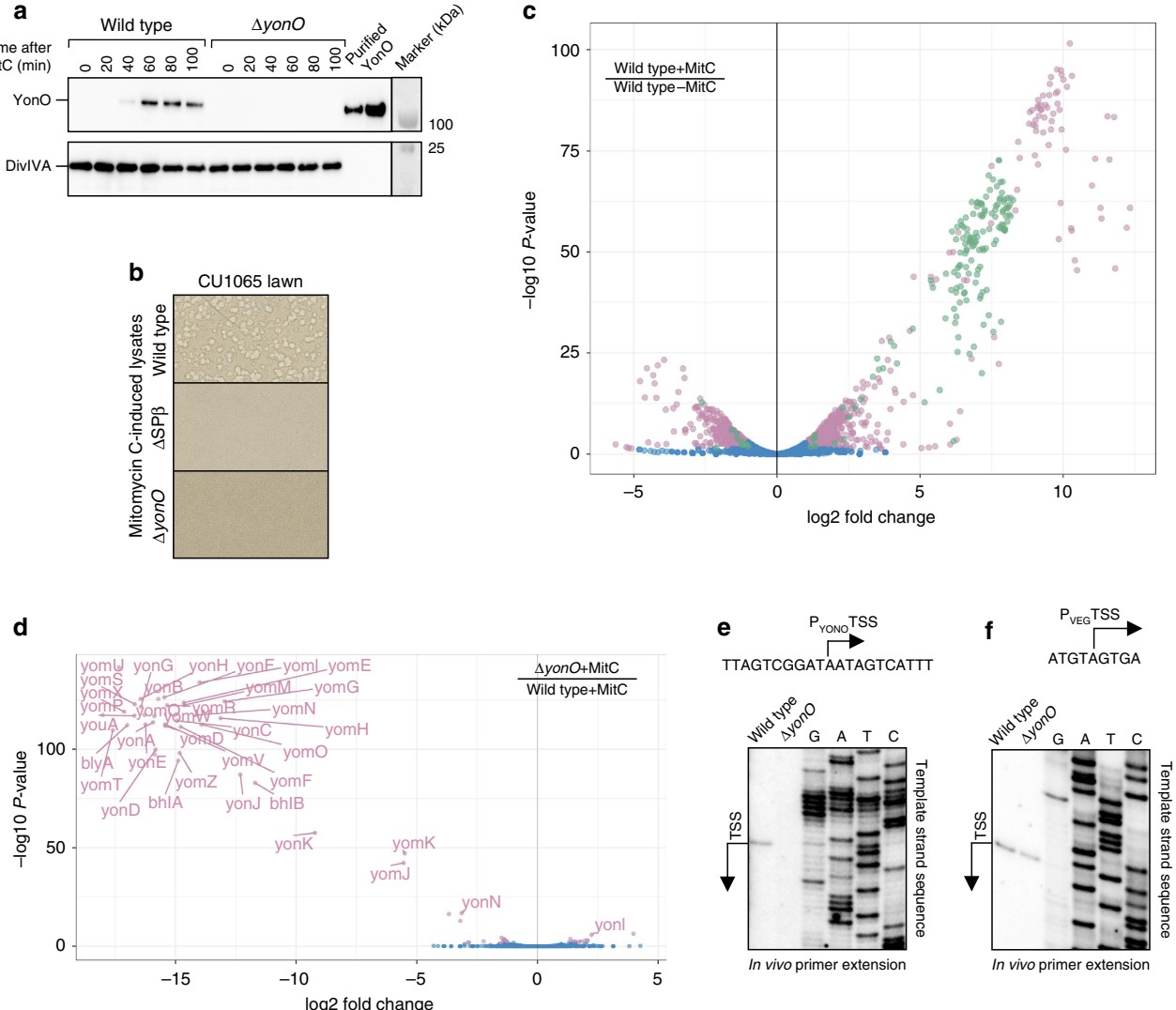

**Figure 2 | YonO is a *bona fide* RNAP of SPβ.** (**a**) Western blot analysis of YonO expression upon SPβ induction with mitomycin C. DivIVA, a constitutively expressed cell division protein, was probed as a loading control. (**b**) Plaque formation assay with SPβ sensitive strain (CU1065) infected with lysates of mitomycin C (MitC)-induced wild type, ΔSPβ and ΔyonO *B. subtilis* strains. Parts of agar plates are shown. (**c**) A volcano plot of the gene expression fold change against *P*-value for the 4314 genes between WT strains with and without mitomycin C treatment. Genes were considered significant at a fold change of 2 and a *P*-value threshold of 0.05, following correction using the Benjamini–Hochberg false discovery rate. SPβ genes with statistically significant changes in expression are shown in green (*N* = 157) with other significant genes in pink (*N* = 1,072). Blue indicates all the genes (out of 4,314) without a statistically significant change in expression. (**d**) A volcano plot of the gene expression fold change against *P*-value for the 4,314 genes between wild-type and ΔyonO strains upon prophage induction. Criteria for significant change are as in panel **c**. Out of the 148 SPβ prophage genes induced by mitomycin C (see panel **c**), 37 were not expressed in the ΔyonO strain (pink dots with labels; see also Supplementary Fig. 2). Blue dots indicate all the genes (out of 4,314) without a statistically significant change in expression. (**e**) Confirmation of the transcription start site (TSS) of the promoter used by YonO ($P_{YONO}$) to transcribe the operon of SPβ late genes, as determined by RNA-seq, using primer extension. (**f**) Deletion of YonO does not affect transcription from constitutive $P_{VEG}$ promoter of *B. subtilis* as confirmed by primer extension.

pausing[30]. The β lobe participates in open complex formation and is required at lower temperatures[31,32]. Interestingly, however, when tested in mild conditions *in vitro*, all these domains appear to be dispensable for efficient RNA synthesis by msRNAP; they only become essential in the context of the regulation of transcription. Accordingly, we found that YonO is less responsive to transcription pausing, which is a main mechanism for regulation of elongation by msRNAPs, and which is controlled by some of the above domains. The apparent absence of such fine-tuning in YonO suggests that these mechanisms emerged later during the evolution of the msRNAPs.

## Methods

**Cloning.** All cloning and DNA manipulations were performed using standard techniques. All oligonucleotides were obtained from Integrated DNA Technologies. Sequences for the oligonucleotides used in this work are listed in Supplementary Table 1. Plasmid constructs were verified by sequencing with GATC Biotech. *B. subtilis* strains used in the study were wild-type strain, 168ca (*trpC2*); strain deleted for *yonO*, ΔyonO (*trpC2 ΔyonO* this work); strain deleted for SPβ prophage, ΔSPβ (*trpC2 ΔSPβ*); and a strain sensitive to infection by SPβ phage, CU1065 (*trpC2* SPβ^S)[33–35].

**Purification of YonO.** *yonO* was cloned into the pET-28a expression vector, and YonO was expressed in *E. coli* T7 Express cells (New England Biolabs) with an N-terminal 6xHis-tag. To obtain YonO[3D > 3N], the *yonO* pET-28a construct was subjected to site-directed mutagenesis using Quikchange XL II (Agilent). Cells were

grown in LB at 37 °C, and protein expression was induced at $OD_{600}$ of 0.4 with 1 mM isopropyl-β-D-1-thiogalactoside (IPTG) at 18 °C for 16 h. Harvested cells were re-suspended in grinding buffer (50 mM Tris-HCl pH 7.9, 200 mM NaCl, EDTA-free protease inhibitor cocktail (Roche)) and disrupted by sonication. The lysate was clarified by centrifugation and made to 20 mM imidazole before being loaded on a His Trap HP column (GE Healthcare) pre-equilibrated with 20 mM Tris-HCl pH 7.9 and 600 mM NaCl. Protein eluted in 100 mM imidazole was bound to a HiTrap Heparin column (GE Healthcare) equilibrated in 10 mM Tris-HCl pH 7.9 and 600 mM NaCl, and eluted by a gradient increase of NaCl concentration to 1 M. Fractions containing YonO were concentrated and further purified on a Superdex 200 16/60 column equilibrated in 50 mM Tris-HCl pH 7.9 and 500 mM NaCl. Purified YonO was concentrated and dialysed overnight into storage buffer (20 mM Tris-HCl pH 7.9, 50% glycerol, 200 mM KCl, 1 mM dithiothreitol and 0.1 mM EDTA) and stored at − 20 °C.

**Transcription in vitro.** In vitro transcription reactions were performed using assembled elongation complexes[11,15,18]. For sequences of DNA and RNA oligonucleotides, see sequences in figures and Supplementary Table 2. Ten microlitre reactions containing 0.5 pmol of RNA were incubated with 1 pmol of template strand DNA for 5 min, and 5 pmol of wild type or mutant YonO or E. coli core msRNAP were added for another 5 min. Where indicated, 5 pmol of non-template was added and complexes were incubated for a further 5 min. Five microlitres of Ni-NTA agarose beads (GE Healthcare) equilibrated in transcription buffer (20 mM Tris-HCl pH 8.5, 40 mM KCl and 10 mM $MgCl_2$) were added for 5 min with gentle shaking. Beads were washed with transcription buffer (with or without 1 M KCl). For RNA hydrolysis reactions, complexes were additionally washed with transcription buffer of the relevant pH lacking $Mg^{2+}$. With the exception of RNA14, all RNA and DNA primers were labelled at the 5′ end with γ-[$^{32}$P]-ATP and T4 polynucleotide kinase (Fermentas). RNA14 was labelled at the 3′ end by incorporation of α-[$^{32}$P]-GMP in the assembled elongation complex with subsequent washing. Reactions were initiated by the addition of NTPs, PPi (concentrations specified in figures) and/or 10 mM $MgCl_2$ or $MnCl_2$ and allowed to proceed at 37 °C for 30 min or times indicated in figures. Where specified, 5 units of RNase H (New England Biolabs) were added to reactions for 20 min at 37 °C. All reactions were stopped by the addition of formamide containing buffer. Products were separated by 24% denaturing Urea-PAGE and revealed by phosphorimaging (GE Healthcare). To calculate $K_m$ values, reaction rates obtained from a range of substrate concentrations were fitted to the Michaelis–Menten equation using SigmaPlot software ( ± in the text is s.d. from at least three independent experiments). All experiments were repeated at least three times.

**Affinity purification of α-YonO polyclonal antibodies.** YonO was purified as before with the exception that 20 mM HEPES KOH pH 7.9 replaced Tris-HCl in all purification buffers. Four milligrams of YonO were coupled to 0.8 g CNBr-activated sepharose (GE Healthcare) following the manufacturer's instructions.

Rabbit serum raised against YonO cleaved of its N-terminal 6xHis tag was obtained from Eurogentec (Belgium). Purification of α-YonO antibodies was performed using the YonO coupled sepharose, following the protocol detailed in Banzhaf et al.[36]. All steps were performed at 4 °C. YonO-coupled sepharose was washed with 25 ml coupling buffer (100 mM $NaHCO_3$ pH 8.3, 10 mM $MgCl_2$, 500 mM NaCl and 0.1% Triton X-100) and blocked overnight by incubation with 10 ml blocking buffer (200 mM Tris-HCl pH 8, 10 mM $MgCl_2$, 500 mM NaCl and 0.1% Triton X-100). YonO-coupled sepharose was washed with 20 ml acetate buffer (100 mM NaOAc pH 4.8, 10 mM $MgCl_2$, 500 mM NaCl and 0.1% Triton X-100) followed by 20 ml blocking buffer. These washing steps were repeated three times. Next, YonO-coupled sepharose was washed with 10 ml of elution buffer I (100 mM glycine pH 2 and 0.1% Triton X-100) followed by 30 ml of buffer I (10 mM Tris-HCl pH 7.2, 10 mM $MgCl_2$, 1 M NaCl and 0.1% Triton X-100).

Ten millilitres of rabbit serum were diluted with 35 ml of serum buffer (10 mM Tris-HCl pH 7.4 and 0.1% Triton X-100). The diluted serum was clarified by centrifugation (10 min at 8,000g) and the supernatant was incubated for 20 h with YonO-coupled sepharose. The YonO-coupled sepharose was allowed to settle in a 10 ml gravity flow column (Bio-Rad) while diluted serum was allowed to flow through the column. The column was washed with 20 ml buffer I and buffer II (10 mM Tris-HCl pH 7.2, 10 mM $MgCl_2$, 150 mM NaCl and 0.1% Triton X-100). Bound antibodies were eluted with six 1 ml aliquots of elution buffer I. Each aliquot was eluted into 200 µl elution buffer II (2 M Tris HCl pH 8). Antibodies were visualized by SDS PAGE. Fractions containing antibodies were made to 20% glycerol and stored at − 80 °C.

**Construction of B. subtilis ΔyonO strain.** The marker-free (scarless) B. subtilis ΔyonO strain was constructed using the method as described in Morimoto et al.[37]. This method utilizes genomic integration of a MazF selection cassette featuring an IPTG inducible E.coli mazF gene and a spectinomycin resistance marker. Recombinant PCR steps were used to generate a large DNA fragment consisting of, in the order listed, Fragment A, Fragment B, MazF cassette and Fragment C. Fragment A, B and C refers to the 1,000 bp DNA sequences immediately upstream

of yonO, immediately downstream of yonO and within the yonO coding sequence, respectively. This PCR fragment was transformed into B. subtilis 168ca strain and selected for on LB agar containing 100 µg ml$^{-1}$ spectinomycin. To select cells that have lost the MazF cassette and yonO via intra-molecular recombination between Fragment B and the genomic DNA immediately downstream of yonO, cells were grown to exponential phase in LB and plated onto LB agar containing 1 mM IPTG. The deletion of yonO was confirmed through PCR and sequencing.

**Induction of SPβ prophage and lysate preparation.** The following protocol is adapted from Harwood and Cutting,[38]. Overnight cultures of the relevant B. subtilis strain were grown at 30 °C in MMB medium (1% bacto-tryptone, 0.5% bacto-yeast extract, 1% NaCl, 5 mM $MgCl_2$ and 0.1 mM $MnCl_2$). These cultures were used to inoculate fresh MMB medium to an $OD_{600}$ of 0.02. The cultures were grown at 37 °C until they reached mid-log phase ($OD_{600} \sim 0.5$), at which point mitomycin C (Melford) was added to a final concentration of 0.5 µg ml$^{-1}$. Cell cultures were grown at 37 °C with shaking for a further 2 h at which point lysis occurred. Lysis occurred in all strains, including ΔyonO and ΔSPβ, due to mitomycin C-dependant induction of non-SPβ lytic enzymes encoded elsewhere in the B. subtilis genome.

Cell lysates were centrifuged (20,000g, 2 min) and supernatant filtered through a 0.45 µm PVDF filter (Merck-Millipore). Strain CU1065 (SPβ sensitive-SPβ$^S$) was grown until 0.5 $OD_{600}$. Three hundred microlitres of CU1065 culture were incubated at room temperature with 100 µl of lysate containing SPβ particles for 2 min. MMB overlay agar (MMB media supplemented with 0.5% agar) was added and cells were plated onto MMB bottom agar (MMB media with 2% agar). Plates were incubated overnight at 37 °C.

**Western blot.** Cultures were grown and SPβ was induced as described above. After induction, cells were harvested at 20 min intervals, re-suspended in 10 volumes of grinding buffer and disrupted by sonication. Lysates were cleared by centrifugation. The protein concentration of the lysates was determined using Bradford reagent (Bio-Rad) to allow for equal loading of samples (5 µg of total protein per lane). Proteins were resolved by electrophoresis in 4–20% gradient SDS gel (Expedeon). Proteins were blotted onto Hybond-P PVDF membrane (GE Healthcare) using wet transfer apparatus (Bio-Rad) and probed using α-YonO polyclonal primary antibodies (described above). Primary antibodies were diluted 1 in 1,000 before use and incubated with the blot for 1 h at 4 °C. Polyclonal horseradish peroxidase-conjugated secondary antibodies produced in goat (Sigma-Aldrich. Catalogue number A0545) were diluted 1 in 10,000 and incubated as before. The western blot was visualized using ECL plus substrate (Thermo Scientific, Catalogue number 32132) and the ImageQuant LAS 4,000 mini digital imaging system (GE Healthcare). Uncropped blots can be seen in Supplementary Fig. 3.

**Transcriptome sequencing (RNA-seq).** Fresh MMB media was inoculated to 0.02 $OD_{600}$ with an overnight culture of B. subtilis grown in MMB media at 30 °C. Cultures were grown at 37 °C until 0.5 $OD_{600}$ was reached, at which point the cultures were halved. In one half, SPβ was induced by the addition of mitomycin C to a final concentration of 0.5 µg ml$^{-1}$. Sixty minutes post induction, cultures were harvested by centrifugation at 20,000g and the pellets were flash frozen.

Cells were disrupted using the FastRNA Pro Blue Kit (MP Bio). Briefly, cell pellets were resuspended in 1 ml of the supplied phenol lysis solution and added to the Lysing Matrix before disruption in a Precellys 24 homogenizer (Bertin Technologies). After centrifugation, the supernatant was removed and chloroform extracted. Total RNA was then isolated from the aqueous phase using the Total RNA Purification Plus Kit (Norgen). Prior to isolation following the manufacturer's instructions, an equal amount of the provided lysis solution was added to the aqueous phase from the chloroform extraction.

Total RNA quality and concentration was determined using a Bioanalyser 2100 with an RNA 6000 nano chip (Agilent) according to the manufacturer's instructions. Sample preparation including rRNA depletion, Terminator 5′-Phosphate-Dependent Exonuclease treatment, library construction and sequencing was performed by Vertis Biotechnologie AG (Germany) as described in Sharma et al.[39], with the exception that sequencing was performed on an Illumina NextSeq 500 (ref. 39). Samples were prepared in biological triplicate.

**Primer extension.** Five micrograms of total RNA (purified as described above) and 2 pmol of 5′ γ-[$^{32}$P]-ATP labelled primer were used for primer extension with Thermoscript reverse transcriptase according to the manufacturer's instructions (Invitrogen). cDNA and control DNA sequencing reactions were resolved by 6% denaturing sequencing Urea-PAGE and revealed by phosphorimaging (GE Healthcare). Primer sequences used for the primer extension and to generate the PCR fragments for sequencing reactions are listed in Supplementary Table 1.

**RNA-seq data analysis.** Sequence quality was assessed using FastQC (http://www.bioinformatics.babraham.ac.uk/projects/fastqc) and adaptors

removed where necessary. Reads were pre-processed using FASTX-toolkit (http://hannonlab.cshl.edu/fastx_toolkit/index.html) by trimming to a maximum length of 45 nt, before trimming of the low-quality sequence from the 3′ end using a Phred threshold of 20. The processed reads were aligned to the *B. subtilis* reference genome (NCBI accession: NC_000964) using Bowtie[40], allowing only unique alignments with up to three mismatches, before conversion to bam format using samtools[41]. Gene coverage was calculated using the Rsubread package[42] and differentially expressed genes identified using the EdgeR exact test[43]. Genes were considered significant at a fold change of 2 and a *P*-value threshold of 0.05, following correction using Benjamini–Hochberg false discovery rate[44]. Per base coverage was calculated using BEDtools[45]. Counts were normalized to the number of reads in each library, and then multiplied by the median number of reads across all libraries to restore the data range. Transcription start sites were identified using TSSpredator[46] using 90th percentile normalization.

**Data availability.** RNA-seq data that support this study have been deposited in NCBI GEO, which are accessible with the accession number GSE80786. The following NCBI gene accession codes were used in this work: NC_000964. All the other data are available from the corresponding authors upon reasonable request.

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

## Acknowledgements

We thank Pamela Gamba and Heath Murray for help with *B. subtilis* genetics and Simon J. Cockell for help with the bioinformatics. This work was supported by Wellcome Trust Senior Investigator Award (102851), grants from the Leverhulme Trust (PLP-2014-229), the UK Medical Research Council, and the UK Biotechnology and Biological Sciences Research Council to N.Z., and Royal Society University Research Fellowship to Y.Y.

## Author contributions

D.F. performed experiments, K.J. performed bioinformatic analysis, Y.Y. and N.Z. conceived and conducted the study, and N.Z. wrote the paper.

## Additional information

**Competing interests:** The authors declare no competing financial interests.

**Publisher's note**: 

