## [Peer Review File · Nature Communications]

Reviewers' comments:

Reviewer #1 (Remarks to the Author):

Previous bioinformatic, sequence and structural analyses had revealed the presence of open reading frames in the genomes of prophages of some firmicutes, the genomes of some firmicutes and cyanobacteria distantly related to the multisubunit (5 subunit in bacteria) DNA-dependent RNA polymerases (msRNAP) and eukaryotic RNA-dependent RNA polymerases responsible for post-transcriptional gene silencing, encoding for single subunit proteins.

Forrest et al describe the purification and characterization of one such protein, the product of the YonO gene encoded by the *B. subtilis* prophage SP β . The sequence homology of YonO to msRNAPs is restricted to few catalytically-essential residues on the two double-psi- β -barrel domains of β and β' that constitute the active site. The characterization of the protein activity follows well-tested approaches.

- 1- The protein is active for elongation when loaded onto a classical elongation scaffold.
- 2- It is a DNA-dependent RNAP, with marked preference for a DNA template and NTPs to yield an RNA transcript
- 3- Elongation complexes are salt-stable when containing an 8 bp RNA:DNA hybrid
- 4- Addition of the 4 NTPs led to run off transcription
- 5- Has high preference for DNA template and rNTP substrates
- 6- Is required for transcription of the *B. subtilis* SP β late genes after induction

Concerns

- 1- Those results raised the following question: "It walks as a duck, quacks as a duck but is it really a duck?" The authors point out to the residues with homology to β and β' of msRNAPs required for catalysis. Those should be mutated, the proteins purified and tested for activity.
- 2- It is surprising that, although not sequence related, the authors do not refer to the simple single subunit RNA polymerases, which lack proofreading and back-tracking activity, show reduced pausing, are faster than the ms RNAPs and quite processive. A priori, a trigger loop is not essential
- 3- One outstanding question is the mechanism of promoter recognition. The authors indicate that those are questions for the future. Having a single promoter, footprinting, template base changes and crosslinking to identify protein and template residues recognized upon binding is very straightforward.

Reviewer #2 (Remarks to the Author):

The cellular DNA-dependent RNA polymerases are universally multi-subunit enzymes. Thus, the conjecture, more than 10 years ago, that an ORF of a *B. subtilis* inducible prophage might point to a very ancient (pre-LUCA) single-subunit origin of the contemporary multi-subunit enzymes should have aroused wider interest and follow-up than was the case. This interesting initial communication takes up the challenge of that conjecture by showing that the YonO protein of the *B. subtilis* 168 SP β prophage is a DNA-dependent RNA polymerase and that its function is essential for expression of viral late genes after induction of the lytic viral multiplication cycle. In my judgment, this brief note and the promise of further work on this very special single-subunit RNAP will elicit sufficiently wide interest to merit publication in Nature Communications.

I do have specific concerns to convey to the authors, but all of them involve further experiments that will require little time. In text order (item 5 is my principal concern):

- 1) line 75 and Fig 1c: Is the YonO protein "strict" with Mn(+2) in place of Mg(+2)? I suspect that perhaps not.

2) line 80 and Fig 1d (also line 173 and Fig. 2e): Is the transcript released or is an RNA-DNA hybrid formed (tested with RNases H,A)?

3) line 84 and Supp. Fig 1: Is YonO prone to non-templated 3'-end addition? It looks like that might be the case. Also, if the effective K_d for NTPs is less than 1 μM (as it might be in view of Fig. 1d) are there concerns about the purity of the CTP and UTP, used at high concentrations for the misincorporation test?

4) line 89 and Fig. 1e: I don't think that this expt. at a single pH justifies the absolute statement. The characteristic pKs of the hydrolytic process of the 2 enzymes might be different. A pH titration is indicated.

5) line 132 and Fig. 2e: The duplex DNA transcription expt. is inadequate (besides being a bit of a mess): What establishes that the thick band in the left-hand lane is a single run-off transcript (cf Fig. 2d) and, more importantly, that it is not an end-to-unknown-terminator transcript? And is it released from the template or is it produced as an RNA-DNA duplex? Hasn't the 5' end of the in vitro transcript already been mapped along with the in vivo RNA? The implication of this experiment that the authors seem to convey (see the cartoon at the top of Fig. 2e and "run-off transcript" at the side) is that the YonO protein, like the unrelated T7 single-subunit RNA polymerase, finds its transcriptional start site without the aid of a sigma-like (or archeal TBP/TFB-like) initiation factor. A better and better-executed experiment is called for to even suggest this important implication. The last sentence of Discussion states that work on initiation of transcription is continuing. In Fig. 2e, the authors imply they believe they already know the answer, but the evidence is just not there. I hope it can be provided in this publication. If time constraints do not allow that, Fig. 2e needs to be omitted (which would be regrettable).

(signed)

E. Peter Geiduschek

Reviewer #3 (Remarks to the Author):

This paper reports a significant discovery, an RNA polymerase with apparent homology to the universal family of multisubunit RNA polymerases, but of a highly limited and restricted sort: not much more than a handful of essential residues in the active center, in a single polypeptide. Nonetheless, the relation is persuasive and evidence of the activity and function of the RNA polymerase are convincing. The data are generally good and technically well produced. The discovery is important enough to deserve a publication of some prominence.

Several questions arise.

1. No comment is made about the +/- NT components of Fig. 1B.
2. It is argued that the YonO polymerase pauses much less than the E. coli multisubunit RNA polymerase. However, the experiment was performed at only one (low) concentration of substrate. Wouldn't this be the result if the K_m were simply quite different for the two enzymes, but the pausing behavior not significantly different?
3. The identification of the promoter by runoff synthesis in Fig. 2E does show a (fuzzy) band of the expected size, but there is other RNA as well. It would have been useful to show that random fragments make no bands of this size, or that different fragments with the sequence make runoffs of predicted size. This comment has partly to do with the absence of more precise description of how the triphosphate end analysis was done to make it the expectation that there is only one 5' end that should map in this fragment. Probably the authors can satisfy this with a few more words.

4. Could the authors reference the non-essentiality of the various domains of multisubunit RNA polymerases, as asserted in the discussion (p. 8). Or would these be the references already given for the functions of these domains?

Reviewers' comments:

Reviewer #1 (Remarks to the Author):

Previous bioinformatic, sequence and structural analyses had revealed the presence of open reading frames in the genomes of prophages of some firmicutes, the genomes of some firmicutes and cyanobacteria distantly related to the multisubunit (5 subunit in bacteria) DNA-dependent RNA polymerases (msRNAP) and eukaryotic RNA-dependent RNA polymerases responsible for post-transcriptional gene silencing, encoding for single subunit proteins.

Forrest et al describe the purification and characterization of one such protein, the product of the YonO gene encoded by the B. subtilis prophage SP β . The sequence homology of YonO to msRNAPs is restricted to few catalytically-essential residues on the two double-psi- β -barrel domains of β and β' that constitute the active site. The characterization of the protein activity follows well-tested approaches.

1- The protein is active for elongation when loaded onto a classical elongation scaffold.

2- It is a DNA-dependent RNAP, with marked preference for a DNA template and NTPs to yield an RNA transcript

3- Elongation complexes are salt-stable when containing an 8 bp RNA:DNA hybrid

4- Addition of the 4 NTPs led to run off transcription

5- Has high preference for DNA template and rNTP substrates

6- Is required for transcription of the B. subtilis SP β late genes after induction

Concerns

1- Those results raised the following question: "It walks as a duck, quacks as a duck but is it really a duck?" The authors point out to the residues with homology to β and β' of msRNAPs required for catalysis. Those should be mutated, the proteins purified and tested for activity.

As suggested by the Reviewer, we mutated YonO's aspartate triad that chelates the first Mg²⁺ ion in msRNAPs, and that is among the few amino acids conserved between YonO and msRNAPs. We purified and characterised the mutant enzyme, which was absolutely inactive in RNA synthesis, despite forming stable ECs (Fig. 1c). The result indicates that the aspartate triad are indeed the catalytic residues of YonO.

2- It is surprising that, although not sequence related, the authors do not refer to the simple single subunit RNA polymerases, which lack proofreading and back-tracking activity, show reduced pausing, are faster than the ms RNAPs and quite processive. A priori, a trigger loop is not essential

We agree with the Reviewer that the higher processivity, lack of proofreading, etc., could be a result of functional convergence between evolutionary unrelated bacteriophage RNAPs (single-subunit RNA polymerases and YonO), in contrast to msRNAPs, because of their similar tasks during phage development. We have added this discussion to the revised version.

However, we refrain from discussing the essentiality of the Trigger Loop for the following reason. As mentioned by the Reviewer, the single-subunit RNA polymerases (ssRNAPs) are not related to msRNAPs or YonO. Therefore, while the convergence of the functions of RNA synthesis between ssRNAPs and msRNAPs is highly remarkable indeed, the structural and/or chemical aspects of these functions cannot be compared directly. The O-helix of ssRNAP was proposed to be functionally analogous to the Trigger Loop in delivering the substrate to the active conformation. However,

although it suggests the importance of the rearrangement of the active centre during NMP addition, this analogy is limited. For example, the catalysis of phosphodiester bond formation by ssRNAP involves acid-base catalysis, which is not the case for msRNAP, whose Trigger Loop participates in stabilisation of the transition state of the reaction. Thus one cannot argue on the essentiality or non-essentiality of the Trigger Loop based on comparisons between msRNAP and ssRNAP.

In contrast, the properties of YonO and msRNAPs, such as processivity, proofreading, etc., can be directly compared as these enzymes share a common ancestor. Hence, it is surprising that an enzyme apparently lacking the part of the active centre conserved among all its relatives (msRNAPs) is as active as them.

3- One outstanding question is the mechanism of promoter recognition. The authors indicate that those are questions for the future. Having a single promoter, footprinting, template base changes and crosslinking to identify protein and template residues recognized upon binding is very straightforward.

As suggested by Reviewer 2, because of the lack of clarity on the mechanisms involved, we have removed panel 2e and accompanying discussions. Experiments on initiation that we performed during revision (footprinting, abortive initiation, etc.) have raised more questions about promoter properties and the requirements for initiation than provided answers. While we can obtain initiation on double-stranded DNA, we do not see formation of the promoter open complex *in vitro* and we cannot explain why *in vitro* transcription by YonO is much weaker than we observe *in vivo*, or why we observe substantial non-specific initiation *in vitro* but not *in vivo*. This may imply a requirement for a specific DNA structure, superhelicity or additional factors to increase specificity and/or efficiency of initiation. Investigation of these mechanisms is not feasible within the revision process, and is a separate, possibly several-year, project that we are starting in the lab.

Reviewer #2 (Remarks to the Author):

The cellular DNA-dependent RNA polymerases are universally multi-subunit enzymes. Thus, the conjecture, more than 10 years ago, that an ORF of a B. subtilis inducible prophage might point to a very ancient (pre-LUCA) single-subunit origin of the contemporary multi-subunit enzymes should have aroused wider interest and follow-up than was the case. This interesting initial communication takes up the challenge of that conjecture by showing that the YonO protein of the B.subtilis 168 SPbeta prophage is a DNA-dependent RNA polymerase and that its function is essential for expression of viral late genes after induction of the lytic viral multiplication cycle. In my judgment, this brief note and the promise of further work on this very special single-subunit RNAP will elicit sufficiently wide interest to merit publication in Nature Communications.

I do have specific concerns to convey to the authors, but all of them involve further experiments that will require little time. In text order (item 5 is my principal concern):

1) line 75 and Fig 1c: Is the YonO protein "strict" with Mn(+2) in place of Mg(+2)? I suspect that perhaps not.

We have repeated the substrate specificity experiment (RNA vs DNA, NTPs vs dNTPs) in the presence of Mg^{2+} or Mn^{2+} . YonO, as well as msRNAPs, can use Mn^{2+} as efficiently as Mg^{2+} . Mn^{2+} had little effect on YonO specificity, apart from the DNA-DNA “hybrid” in the presence of NTPs, where it increased the otherwise poor extension. The results are presented in Supplementary Fig. 1d of the revised version, and are discussed in the text.

2) line 80 and Fig 1d (also line 173 and Fig. 2e): Is the transcript released or is an RNA-DNA hybrid formed (tested with RNases H,A)?

We performed the suggested analysis, and showed that YonO does facilitate RNA disengagement from the template DNA behind the elongation complex, as do msRNAPs (Supplementary Fig. 1c)

3) line 84 and Supp. Fig 1: Is YonO prone to non-templated 3'-end addition? It looks like that might be the case. Also, if the effective K_d for NTPs is less than 1 μM (as it might be in view of Fig. 1d) are there concerns about the purity of the CTP and UTP, used at high concentrations for the misincorporation test?

Indeed, during prolonged incubation in a high NTP concentration, YonO adds non-templated NMPs to the 3' end of the run-off transcript (we found that RNA stays in the complex at the end of template, as in the case of *E. coli* RNAP; Supplementary Fig. 1c). *E. coli* RNAP also added non-templated NMPs but less efficiently. We found that, although the K_m for correct NTPs of YonO was similar or slightly higher than that of *E. coli* RNAP, K_m for incorrect ones was much lower, suggesting a possible explanation for the more efficient 3'-end additions by YonO.

Similarity of K_m s to NTP between two enzymes also suggests that the “faster” transcription by YonO in Fig. 1d (Fig. 1e in revised manuscript) is rather explained by a higher processivity of YonO. To confirm that we performed extension at high (100 μM) NTPs concentration, which showed that *E. coli* msRNAP had still lower processivity as compared to YonO (right panel in Fig. 1e in revised manuscript).

As we found K_m s for the correct NTPs to be similar for YonO and *E. coli* RNAP, the observation of the extension with the wrong NTPs cannot be explained by contamination with correct ones. Furthermore, we have no concerns with the NTPs' purity because in most cases our Urea-PAGE system clearly separates RNAs with different 3' NMPs (this can be seen in the misincorporation gels; Fig. 1f and Supplementary Fig. 1e in the revised version), and, in many cases, transcripts of the same length but with different sequences. Some years ago we did perform additional purification of commercial NTPs (GE Healthcare), but found it be redundant.

4) line 89 and Fig. 1e: I don't think that this expt. at a single pH justifies the absolute statement. The characteristic pKs of the hydrolytic process of the 2 enzymes might be different. A pH titration is indicated.

As suggested by the Reviewer, we performed proofreading reactions at different pHs. As can be seen from Supplementary Fig. 1f of the revised version, while cleavage by *E. coli* RNAP increased significantly with the increase of pH, YonO remained completely inactive even during prolonged incubation. Though we cannot exclude that at an even higher pH YonO may show some activity in

hydrolysis, this can hardly be observed experimentally since at these pHs/times RNA begins to be non-enzymatically hydrolysed in the presence of Mg^{2+} . Similarly, activity with a much higher pKa is unlikely to have biological relevance.

5) line 132 and Fig. 2e: The duplex DNA transcription expt. is inadequate (besides being a bit of a mess): What establishes that the thick band in the left-hand lane is a single run-off transcript (cf Fig. 2d) and, more importantly, that it is not an end-to-unknown-terminator transcript? And is it released from the template or is it produced as an RNA-DNA duplex? Hasn't the 5' end of the in vitro transcript already been mapped along with the in vivo RNA? The implication of this experiment that the authors seem to convey (see the cartoon at the top of Fig. 2e and "run-off transcript" at the side) is that the YonO protein, like the unrelated T7 single-subunit RNA polymerase, finds its transcriptional start site without the aid of a sigma-like (or archeal TBP/TFB-like) initiation factor. A better and better-executed experiment is called for to even suggest this important implication. The last sentence of Discussion states that work on initiation of transcription is continuing. In Fig. 2e, the authors imply they believe they already know the answer, but the evidence is just not there. I hope it can be provided in this publication. If time constraints do not allow that, Fig. 2e needs to be omitted (which would be regrettable).

We performed additional experiments to understand the mechanism of initiation, but these raised more questions than answers. As mentioned in the reply to Reviewer 1, although we can observe run-off transcription, we cannot exclude several nearby starts as the band is usually fuzzy; we do not see formation of the promoter open complex; and cannot explain the lower efficiency and specificity of transcription as compared to our *in vivo* observations. It will not be feasible to address these problems during the revision, and we have removed Fig. 2e and corresponding discussions as suggested by the Reviewer.

Reviewer #3 (Remarks to the Author):

This paper reports a significant discovery, an RNA polymerase with apparent homology to the universal family of multisubunit RNA polymerases, but of a highly limited and restricted sort: not much more than a handful of essential residues in the active center, in a single polypeptide. Nonetheless, the relation is persuasive and evidence of the activity and function of the RNA polymerase are convincing. The data are generally good and technically well produced. The discovery is important enough to deserve a publication of some prominence.

Several questions arise.

1. No comment is made about the +/- NT components of Fig. 1B.

We thank the Reviewer for spotting this miss. We have added a discussion of the experiment. Briefly, while msRNAP elongation complex (EC) without NT is stable at normal ionic strength, it cannot withstand high (1M) salt. This property is explained by the critical role of the downstream DNA duplex in EC stabilisation. In the mentioned experiment, we tried to show similarities of the requirements for EC stability and, thus possibly, their structures for YonO and msRNAP.

2. It is argued that the YonO polymerase pauses much less than the *E. coli* multisubunit RNA polymerase. However, the experiment was performed at only one (low) concentration of substrate. Wouldn't this be the result if the K_m were simply quite different for the two enzymes, but the pausing behavior not significantly different?

We have repeated the experiment at a high (100 μ M) NTP concentration and observed similar behaviour; YonO was more processive than *E. coli* msRNAP (Fig. 1e of the revised version). Furthermore, we measured the K_m for NTP of YonO and *E. coli* msRNAP in elongation complex and found them to be similar.

3. The identification of the promoter by runoff synthesis in Fig. 2E does show a (fuzzy) band of the expected size, but there is other RNA as well. It would have been useful to show that random fragments make no bands of this size, or that different fragments with the sequence make runoffs of predicted size. This comment has partly to do with the absence of more precise description of how the triphosphate end analysis was done to make it the expectation that there is only one 5' end that should map in this fragment. Probably the authors can satisfy this with a few more words.

Triphosphorylated 5' end of the transcript was determined by RNA-seq. In order to enrich for 5' triphosphorylated transcripts during RNA-seq, the RNA sample was treated with 5' terminator exonuclease (TEX), which degrades RNAs containing a 5'-monophosphate, thereby enriching for transcripts containing 5'-triphosphates. We have added this description to the text. However, as suggested by Reviewer 2, we decided to remove figure 2e and the corresponding discussion. The reason for this was that the additional experiments which we have performed (footprinting, abortive synthesis, etc.) raised more questions than delivered answers. We cannot exclude several nearby starts as the band is usually fuzzy (which though is not observed on different DNA templates). We do not yet understand the properties of the promoter, do not observe open complex formation, and cannot explain lower efficiency and specificity of transcription *in vitro* as compared to *in vivo*. Answering these questions is a separate study in the lab, and is not possible within this revision.

4. Could the authors reference the non-essentiality of the various domains of multisubunit RNA polymerases, as asserted in the discussion (p. 8). Or would these be the references already given for the functions of these domains?

The references provided are the references about the functions of these domains. These works showed non-essentiality of these domains in robust RNA synthesis *in vitro*, but their requirements for response to some specific regulatory signals or in some specific conditions.

REVIEWERS' COMMENTS:

Reviewer #1 (Remarks to the Author):

The authors have answered many of the reviewers' requests.

The presence of far homologs of β and β' in a single polypeptide is not surprising. It has been shown that the E. coli RNAP is active when the β and β' subunits are fused. The authors have confirmed that the D-triad indeed is required for catalysis and other properties have been clarified. Was looking forward to the finding of the determinants of specificity that, given a single promoter, would have been straightforward!!! It is unfortunate that this most interesting property, the specificity of initiation as described in the original manuscript, cannot be ascertained at this time. Alas, does it need an additional factor, is it encoded in the 100kDa polypeptide?

Lucia B. Rothman-Denes

Reviewer #2 (Remarks to the Author):

The authors have responded to comments from all reviewers, with additional experiments and corresponding text changes. A flawed preliminary experiment dealing with the crucial question of whether the Yon O RNAP recognizes promoters autonomously has been appropriately withdrawn. Leaving that issue for the future is disappointing, of course, but the revised, improved ms. retains considerable interest and novelty and is, in my judgement ready for publication. A re-reading of the text leads me to suggest a very small number of minor text changes:

- l.105: delete "can"

- l.109: replace "hybrid" with "duplex"?

- l. 108: ...Mn⁺² stimulated extension relative to Mg⁺²...?

l. 136: specify the higher pH?

l. 201: ...that need to synthesize a large...?

l.524: change "with a statistically insignificant" to "without a significantly significant" (statistically proving insignificance is a tall order!)

Reviewer #3 (Remarks to the Author):

I think that the authors have answered the reviewers' questions, including mine, quite thoughtfully. Many issues were clarified by additional experiments or explanations. Clearly the nature of the initiation complex will remain mysterious until further work is done, but the publication is fine as is.

Please see below our response to the Reviewers comments (our response is in red).

REVIEWERS' COMMENTS:

Reviewer #1 (Remarks to the Author):

The authors have answered many of the reviewers' requests.

The presence of far homologs of β and β' in a single polypeptide is not surprising. It has been shown that the E. coli RNAP is active when the β and β' subunits are fused. The authors have confirmed that the D-triad indeed is required for catalysis and other properties have been clarified. Was looking forward to the finding of the determinants of specificity that, given a single promoter, would have been straightforward!!! It is unfortunate that this most interesting property, the specificity of initiation as described in the original manuscript, cannot be ascertained at this time. Alas, does it need an additional factor, is it encoded in the 100kDa polypeptide?

Lucia B. Rothman-Denes

We have rewritten the end of the last Results paragraph to reflect that we currently don't know the mechanism of initiation, and if YonO recognises promoters on its own or requires any additional factors (the changes are tracked in the submitted Word file).

Reviewer #2 (Remarks to the Author):

The authors have responded to comments from all reviewers, with additional experiments and corresponding text changes. A flawed preliminary experiment dealing with the crucial question of whether the Yon O RNAP recognizes promoters autonomously has been appropriately withdrawn. Leaving that issue for the future is disappointing, of course, but the revised, improved ms. retains considerable interest and novelty and is, in my judgement ready for publication. A re-reading of the text leads me to suggest a very small number of minor text changes:

- l.105: delete "can"

- l.109: replace "hybrid" with "duplex"?

- l. 108: ...Mn²⁺ stimulated extension relative to Mg²⁺...?

l. 136: specify the higher pH?

l. 201: ...that need to synthesize a large...?

l.524: change "with a statistically insignificant" to "without a significantly significant" (statistically proving insignificance is a tall order!)

We have incorporated all the suggested changes (the changes are tracked in the submitted Word file).

Reviewer #3 (Remarks to the Author):

I think that the authors have answered the reviewers' questions, including mine, quite thoughtfully. Many issues were clarified by additional experiments or explanations. Clearly the nature of the initiation complex will remain mysterious until further work is done, but the publication is fine as is.

Reviewer does not have any concerns.